

# Vertical seismic profiling with distributed acoustic sensing images the Rotliegend geothermal reservoir in the North German Basin down to 4.2 km depth

Jan Henninges[1], Evgeniia Martuganova[1], Manfred Stiller[1], Ben Norden[1], Charlotte M. Krawczyk[1,2]

[1] Helmholtz Centre Potsdam, GFZ German Research Centre for Geosciences, Potsdam, 14473, Germany
[2] Technische Universität Berlin, Berlin, 10587, Germany

*Correspondence to*: Jan Henninges (janhen@gfz-potsdam.de)

**Abstract.** We performed vertical seismic profiling at the Groß Schönebeck site in order to gain more detailed information on the structural setting and geometry of the geothermal reservoir, which is comprised of volcanic rocks and sediments of Lower Permian age. During the four-day survey, we acquired data for 61 source positions with the novel method of distributed acoustic sensing (DAS), using hybrid wireline fiber-optic sensor cables deployed in two 4.3 km deep wells. We show that wireline cable tension has a significant effect on data quality. While most of the recorded data has a very good signal-to-noise ratio, individual sections of the profiles are affected by characteristic coherent noise patterns. This ringing noise is a result of how the sensor cable is mechanically coupled to the borehole wall, and it can be suppressed to a large extent using suitable filtering methods. After conversion to strain rate, the DAS data exhibits a high similarity to the vertical component data of a conventional borehole geophone. Upgoing reflections are nevertheless recorded with opposite polarity, which needs to be taken into account during further seismic processing and interpretation. We derived accurate time-depth relationships, interval velocities, and corridor stacks from the recorded data. Based on integration with other well data and geological information, we show that the top of a porous and permeable sandstone interval of the geothermal reservoir can be identified by a positive reflection event. Overall, the sequence of reflection events shows a different character for both wells, which is explained by lateral changes in lithology. The top of the volcanic rocks has a somewhat different seismic response, and no stronger reflection event is obvious at the postulated top of the Carboniferous. The thickness of the volcanic rocks can therefore not be inferred from individual reflection events in the seismic data alone. The DAS method has enabled measurements at elevated temperatures up to 150 °C over extended periods and has led to significant time and cost savings compared to deployment of a conventional geophone chain.

## 1    Introduction

Borehole seismic array measurements benefit from deploying fiber-optic cables and using the novel distributed acoustic sensing (DAS) method. This technique allows for rapid seismic data acquisition because no discrete sensors respectively shifting to multiple levels are necessary (see proofs of concept in e.g. Mestayer et al. 2011; Miller et al. 2012). While issues like the mechanical coupling of the sensor cable and the transfer from strain to geophone-equivalent data are still under discussion (see Hartog et al. 2014; Daley et al. 2016), we used this survey technique and further improved the processing of this new data type for geothermal applications to overcome the classical resolution problem and derive accurate time-depth relationships.



The Groß Schönebeck site is located 40 km N of Berlin in the state of Brandenburg, Germany. It is a research platform operated by the GFZ German Research Centre for Geosciences, which has been set up in order to test if production of geothermal energy from deep-seated reservoirs in the North German Basin is feasible. An enhanced geothermal system (EGS) has been created by hydraulic stimulation of low-permeability sedimentary and volcanic rocks of lower Permian (Rotliegend) age (Huenges et al. 2006; Zimmermann et al. 2010). So far, two deep research boreholes, the former E GrSk 3/90 hydrocarbon exploration well and the Gt GrSk 4/05 geothermal well (referred to as GrSk3 and GrSk4 in the following), exist at the site. For further development of the site, the implementation of a new stimulation concept and drilling of a new well have been proposed (Blöcher et al. 2015).

In order to gain more detailed information on the structural setting and geometry of the reservoir, a 3D seismic survey within an 8 km x 8 km permit area has been carried out in February and March 2017 (Krawczyk et al. 2019). In addition, vertical seismic profiling (VSP) has been performed within two research wells existing at the site. The primary aims of the VSP survey were to establish precise time-depth and velocity profiles, and to image structural elements in the vicinity of the boreholes with higher resolution in three dimensions. A special challenge is the imaging of structures within the reservoir interval of the Rotliegend at 4200 m depth, which is overlain by the 1400 m thick Upper Permian Zechstein salt complex.

The VSP measurement was performed using the novel DAS method. This method is based on optical time-domain reflectometry, and enables to register strain changes along optical sensor cables with high spatial and temporal resolution (Parker et al. 2014). Within recent years, a growing number of VSP surveys has been reported, where the DAS method has successfully been applied using sensor cables permanently installed behind casing or along tubing (e.g. Mestayer et al. 2011; Daley et al. 2013; Götz et al. 2018). This deployment method is very convenient as it allows for data acquisition without well intervention. In cases where such a permanent installation is not possible or has not been performed during construction of the well, a sensor cable can be lowered downhole temporarily, similar to conventional wireline logging. For this wireline deployment method nevertheless only very few experiences exist until now: First tests using an experimental optical wireline logging cable deployed in a 625 m well were described by Hartog et al. (2014), while a more extensive DAS walkaway VSP survey has been performed by Yu et al. (2016) in a vertical well to a depth of 4004 m. Within the current study, we report on the results of a DAS-VSP acquisition on wireline cable to a depth of 4256 m, which to the authors knowledge represents the deepest survey currently documented in literature worldwide.

## 2 Survey design and data acquisition

The target area was defined by the positions of the existing wells, the expected extent of the hydraulic fractures, and the trajectory of the proposed new well. It has a horizontal extent of approx. 700 m x 500 m and a vertical thickness of approx. 300 m. A spiral pattern of 61 source points with offsets between 180 m and 2000 m from the wellheads was chosen, in order to achieve a good 3D coverage of the target area with a uniform distribution of azimuths (Figure 1). Survey planning was based on well trajectories and geometry of the major geologic units (Moeck et al. 2009), taking into account DAS specific acquisition characteristics like directivity and signal-to-noise ratio. The source point positions were optimized based on ray tracing, using average acoustic properties of the major geologic units from a previous regional seismic survey (Bauer et al. 2010). The actual source point locations were then adjusted according to the conditions within the survey area, i.e. location of roads and agricultural areas, as well as required distances to sensible infrastructures like gas lines or buildings.



A listing of the acquisition parameters is contained in Table 1. Energy excitation was performed with four heavy vibrator trucks operating simultaneously at each source position. For acquisition of the DAS data in well E GrSk 3/90 the GFZ hybrid borehole measurement system was used, which allows for deployment of fiber-optic sensors and electric downhole tools in parallel (Henninges et al. 2011). The GrSk3 well is near-vertical (maximum

inclination 7.2°), and the fiber-optic data was acquired to a measured depth (MD) of 4256 m below ground level, which corresponds to a true vertical depth (TVD) of 4245.8 m below ground level (note that all depths in this study are given in MD, if not stated otherwise). Within the well Gt GrSk 4/05, which is deviated up to 49° in the reservoir interval, a second wireline cable containing optical fibers was deployed (maximum DAS acquisition depth 4196 mMD / 4126.1 mTVD). This is an experimental optical wireline cable developed by Schlumberger, referred to as

optical heptacable (Hartog et al. 2014). This cable was also used to deploy a conventional three-component borehole geophone with acceleration characteristics (VSI Versatile Seismic Imager tool, Schlumberger), in order to record several check shots[1] at specific depths within the GrSk4 well. DAS data was acquired on both cables using two separate Schlumberger hDVS (Heterodyne Distributed Vibration Sensing) units.

**Table 1: DAS-VSP acquisition parameters.**

| Parameter | Value |
|---|---|
| Seismic Source | 4 vibrator trucks; Mertz M12 Hemi 48, peak force 200 kN (45100 lbf) each |
| Number of source points | 61 |
| Offset | 188 m - 2036 m |
| Sweep (near offsets) | 10-112 Hz, linear up, 36 s, 360 ms taper |
| Sweep (far offsets) | 10-96 Hz, linear up, 36 s, 360 ms taper |
| Vertical stacking rate | 16 repetitions (nominal) |
| Signal recording | 2 Schlumberger hDVS units, 2 hybrid wireline cables |
| Depth interval receiver channels E GrSk 3/90 | Ground level – 4256 mMD / 4245.8 mTVD |
| Depth interval receiver channels Gt GrSk 4/05 | Ground level – 4196 mMD / 4126.1 mTVD |
| Receiver channel distance (spatial sampling along borehole) | 5 m |
| Gauge length | 20 m |
| Sampling interval | 2 ms |
| Trace length (after correlation) | 4 s |
| Polarity convention | European / EAGE normal |

Fieldwork was carried out within four days from Feb. 15-18, 2017. At the beginning, we performed a start-up test (referred to as day 0 in the following), where suitable source and recording parameters were determined. As a result, we selected a sweep with 10-112 Hz (linear) and 36 s duration for acquisition. For some of the larger offsets, a sweep with reduced frequency range of 10-96 Hz was used. A gauge length of 20 m was selected for online DAS

data processing during recording. This value was later adjusted to 40 m during post-processing as a result of an optimization procedure (see Section 3.1). The DAS measurements were recorded with a temporal sampling of 2 ms and a spatial sampling of 5 m spacing across the entire length of the wells.

During the start-up test, we recorded several shots with variation of the wireline cable tension in the GrSk3 well, in order to test the influence on the mechanical coupling of the cable and the quality of the recorded signals (cf.

Frignet and Hartog, 2014; Constantinou et al., 2016). After the bottom of the drivable depth in well GrSk3 had been reached at 4259 mMD, recordings with increasing amounts of cable slack of 1 m, 5 m, 11 m, and 20 m have been performed. Based on the results, it was decided to keep the wireline cable under almost full tension for recording, as the best overall data quality was found to be achieved under these conditions (see section 4.2).

---

[1] Here and in the following, the term „shot" is used to refer to a single vibroseis record.





Within the following three days (days 1-3), acquisition was performed with a nominal number of 16 repeats for the 61 source positions distributed around the wells (see Figure 1). Nevertheless, due to a technical problem with acquisition in well GrSk4 during day 1, mainly data for well GrSk3 could only be recorded during this time. Therefore, in order to improve the reduced coverage around the GrSk4 well caused hereby, we relocated some of the original source positions from the northern to the southwestern part of the survey area.

## 3 Seismic data processing

As one of the first processing steps, the DAS data recorded along the length of the sensor cables was correlated to the measured depth along the boreholes. This depth correlation was performed using the gamma-ray logs recorded during running in hole with the sensor cables, as well as travel time data from check shots recorded at 1200 m, 2400 m, 3600 m, and 4207 m depth in the GrSk4 well. During further processing, the depths were transferred to vertical depths below the seismic reference datum, which is mean sea level (TVDSS, True Vertical Depth Sub Sea), using the geometries of the borehole trajectories.

### 3.1 Gauge length optimization

The choice of an optimized gauge length value is an essential part of the DAS data acquisition and processing. This parameter has a significant effect on the signal-to-noise ratio of the data and on resolution in the frequency domain. Dean et al. (2017) presented an approach, which helps to maximize the signal-to-noise ratio, while keeping interfering influences on the frequency content below a desired threshold value. By selecting an optimum gauge length $GL_{opt}$, a favorable compromise between these two factors can be achieved, using

$$GL_{opt} = \frac{Rv}{f_p} \tag{1}$$

with $R$ the gauge length / spatial wavelength ratio, $v$ acoustic velocity, and $f_p$ peak frequency.

The graphs presented in Figure 2 show the dependence of signal-to-noise ratio and resulting wavelength on $R$ for the conditions of the current survey. According to this, optimum conditions within the desired limits are found for $R$-values between 0.46 and 0.56. For an intermediate $R$ value of 0.5, an optimum gauge length of 39 m is calculated using (1), for a velocity of 4800 m/s, which has been extracted from the interval velocities derived for the Rotliegend reservoir interval (see section 4.3), and a middle frequency of 61 Hz for the 10-112 Hz sweep. Therefore, the acquired DAS-VSP data was reprocessed accordingly, using the derived optimum gauge length value.

### 3.2 Pre-processing

An overview of the further seismic data processing steps is contained in Table 2. Seismic pre-processing included stacking and correlation with the pilot sweep. The hDVS output strain data was then transformed to strain rate by differentiation in time, resulting in a 90° phase shift. The strain-rate data is proportional to acceleration (Daley et al. 2016), and acceleration is in phase with the pilot sweep (Sallas 1984).





**Table 2: Sequence of processing steps for zero-offset DAS-VSP data sets.**

| Processing step | Methods, parameters, and description |
|---|---|
| *Pre-processing* | Diversity stack of shots (suppression of impulsive noise) |
| | Correlation with pilot sweep |
| | Conversion to strain rate (time derivative) |
| First arrival time picking | Peak of direct downgoing wave |
| *Interval velocities* | Correct times to vertical |
| | Velocity inversion of travel time data |
| Data preconditioning | Amplitude corrections (spherical divergence compensation and lateral balancing) |
| | Coherent (ringing) noise suppression (Burg adaptive deconvolution) |
| Wavefield separation | Subtraction of downgoing P-wave field (median filter) |
| Waveshaping / zero-phasing of upgoing wavefield, removal of multiples | Deterministic deconvolution, using operator derived from downgoing wavefield |
| Polarity reversal | 180 ° phase shift, to match polarity convention of conventional geophone data |
| *Corridor stack* | Shift to two-way time (horizontal alignment of upgoing reflections), stacking of 0.2 s window after first arrival |

### 3.3 Common-source gathers and coherent noise suppression

Common-source gathers for zero offset, intermediate, and far offset source positions are displayed in Figure 3 and
Figure 4. The common-source gathers are dominated by downgoing P-wave arrivals, and arrivals of upgoing waves
originating from several reflectors at different depths. For several shots, a strong tube wave arriving at later times
is clearly visible as well.

Over several intervals along the wells, a coherent noise with a particular zigzag pattern can be recognized in the
DAS data. Similar noise in DAS data recorded using cables suspended in boreholes has also been described in
some earlier studies, e.g. by Miller et al. (2012), Yu et al. (2016), Cai et al. (2016), and Willis et al. (2019). It has
also been found to occur for tubing-deployed cables, e.g. in the studies of Barberan et al. (2012), and Didraga
(2015).

Several methods for elimination of this "ringing" noise like spectral balancing, deconvolution, and time-frequency
domain filtering (Elboth et al., 2008) were tested. For zero offset data processing we selected to use Burg adaptive
deconvolution (Griffiths et al., 1977). This method is a good compromise between computational effort,
robustness, and application simplicity. A more thorough description of this ringing noise and further methods of
noise suppression can be found in Martuganova et al. (under review). The filtered data sets are displayed in Figure
3 and Figure 4, together with the unfiltered data sets for comparison.

Usage of self-updating linear prediction operators is the foundation of the Burg adaptive deconvolution method.
The designed filter operator is different at each trace sample. A set of filter coefficients is convolved with the data
in order to predict the future data values at some prediction distance. Coefficient values are re-computed for each
data sample in the seismic record with the criterion of minimising the root-mean-square error. The computations
are performed in forward and reverse direction in the time domain.

The application of Burg adaptive deconvolution resulted in a significant reduction of the coherent ringing noise
(Figure 3 and Figure 4). After filtering, reflections are better visible and sharpened. Nevertheless, not all parts of
the noise can be suppressed, especially in a short time window after the first break arrivals. This residual noise is
difficult to be distinguished from upgoing reflected waves, as the velocity of the noise travelling along the cable
is similar to the compressional velocity of the formation (Martuganova et al., submitted).





## 4    Results and discussion

### 4.1    Comparison of DAS and borehole geophone data

A comparison between the DAS strain rate data and the vertical component of the borehole geophone acceleration data recorded at specific depths in the GrSk4 well is displayed in Figure 5. Note that during recording of the check-shot data a sweep with 10-88 Hz was used, which is different from the recording of most of the other data during the survey.

The traces recorded at 1200 and 3600 m depth both contain direct P-wave arrivals, at 520 ms and 1090 ms, respectively. The DAS trace for 1200 m depth is strongly influenced by the ringing noise described above, which is confined to a narrow frequency band between 40 and 50 Hz (Figure 5B) at this location. Similar noise characteristics have been observed, e.g. in the study of Chen et al. (2019). This noise is however not evident in the geophone data recorded at the same depth, which suggests that the ringing noise in the DAS data is related to the different deployment methods of the acoustic receivers. While the DAS sensor cable is freely suspended inside the borehole, the geophone tool is clamped to the borehole wall. Closer analysis of the ringing noise shows that the sensor cable acts like a vibrating string within the affected intervals, with resonances occurring at a fundamental frequency and higher overtones (Didraga et al., 2015; Martuganova et al., submitted).

The traces recorded at 3600 m depth (Figure 5E) also contain strong reflected waves, which arrive at around 1190 ms and originate from the base of Zechstein reflectors, at around 3850 m depth (see Figure 4). Overall, the DAS strain rate data exhibits a high similarity with the geophone measurements, except for the upgoing reflections. Here, the DAS strain rate data displays the opposite polarity as the geophone data. This polarity reversal for reflected upgoing waves has also been observed in previous studies, e.g. by Hartog et al. (2014), Mateeva et al. (2014), or Willis et al. (2016). Frignet and Hartog (2014) note that such a polarity flip compared to geophone data is similar to the characteristics of hydrophone sensors.

As a test, we have also converted the DAS data to geophone-equivalent acceleration data, using the method described by Egorov et al. 2018. For this, we performed a transformation of the original DAS strain data into acceleration via filter application in the vertical wavenumber domain ($k_Z$) and further double differentiation in the time domain. The results for the check-shot traces recorded at 2400 m and 3600 m depth are shown in Figure 6. After conversion to acceleration, the DAS data displays the same polarity as the geophone data, also for the upgoing reflections. This is in line with previous results obtained by Correa et al. (2017).

### 4.2    Signal quality

Common-source gathers recorded with different amounts of cable slack in well GrSk3 are displayed in Figure 7. There is a zone with decreased amplitude of the first break signal at the bottom of the well, which increases in length with increasing amount of cable slack. While the random noise is similar, leading to an overall signal-to-noise ratio (SNR) drop within the affected zone, the coherent noise is changing. For the recordings with 1 m, 5 m and 11 m cable slack, a zone with ringing noise is visible at a depth of approx. 2890 mMD. This zone almost disappears in the 20 m cable slack data set, where the zone of decreased first break amplitudes is approximately approaching the same depth. So ringing noise seems to be reduced within the affected zone, likely because of improved mechanical coupling of the cable to the borehole wall. But at the same time, the signal amplitude is significantly reduced within the affected zone as well.





Due to the higher first break amplitudes, the best signal quality overall was assigned to the data set recorded with 1 m cable slack, i.e. under almost full cable tension, and further recording was performed like this. Notably the best seismic record had been found to be recorded under the opposite conditions with released cable tension during

the field trial reported by Frignet and Hartog (2014). Nevertheless, in their study, the optical wireline cable had been deployed in a relatively shallow well of 625 m depth, and the borehole conditions might not be representative for deep wells as in the current study for the Groß Schönebeck case. Constantinou et al. (2016) observed a behavior similar to the current study during a field trial in a well of 2580 m depth at the Rittershoffen site in France. Here, the zone of reduced signal amplitudes was found to coincide with a region where the cable was interpreted to form

a spiral, gradually building up from the bottom of the well when additional cable slack had been introduced. Schilke et al. (2016) investigated the effect of cable slack on the mechanical coupling of a sensor cable deployed in a vertical well using numerical simulations.

For data quality evaluation, the SNR for each trace of the dataset was calculated. The energy of signal and noise was computed as the root mean square (RMS) amplitude within time windows of -10 to +30 ms around the first

arrival and 150 ms at the beginning of the trace before the first arrival, respectively. The signal-to-noise ratio was then calculated in dB using the formula:

$$SNR = 20 log_{10} \frac{RMS_{signal}}{RMS_{noise}} \tag{2}$$

The calculated SNRs are displayed in Figure 8 and Figure 9. The data is sorted for the different acquisition days and with increasing source offset. Each vertical column represents a source location, and the calculated SNR for

each trace is color-coded. Altogether, the data has a good SNR, with average values of approx. 40 dB to 50 dB at a depth region around 1000 m for the smaller offset source locations, decreasing to approx. 4 dB to 10 dB at around 4200 m close to the final depth. There is an overall decrease of the SNR with increasing channel depth and source offset, which corresponds to the decay of signal amplitudes to be expected due to spherical divergence of the acoustic waves. The data for the first acquisition day have similar characteristics for both wells, with slightly larger

SNRs for well GrSk4.

From the start of the second acquisition day, a sharp drop of the SNR is evident in the data recorded in GrSk3 at a depth of approx. 3400 m. In addition, there are further intervals with decreased SNR at depths of approx. 3100 m and 2600-2800 m. Curiously, the SNR for the channels below 3400 m gradually recovers again with increasing depth, until even improved SNRs in comparison to the first acquisition day are reached in the bottom interval.

The observed signal drop at 3400 m after day 1 seems to be similar to the effect of reduced signal amplitudes observed during the slack test. Nevertheless, the configuration of the wireline cable remained unchanged between day 1 and day 2. Accidental introduction of additional cable slack during this time, e.g. by slipping of the wireline winch, or movement of the crane arm holding the cable sheave, can be excluded, as the position of the cables was carefully monitored by placing marks on them after running into the hole. Furthermore, no significant change of

the wireline cable tension at surface has been registered between day 1 and day 2. Other causes must therefore be responsible for the observed effect.

Combined with the remaining coherent noise after filtering, there is a significant heterogeneity in the data, which requires to carefully select the data to be considered during evaluation and interpretation.

### 4.3 Time-depth relationships and interval velocities

For every source point, the travel times of the direct downgoing waves were determined by picking of the first break times (Table 2). A velocity model has been set up based on the geometry of the existing geological model





from Moeck et al. (2009), and by calibrating the model velocities with the picked travel times. Vertical travel times have then been determined by ray tracing through the calibrated model.

For a close to the receiver wells situated zero-offset position, VSP interval velocities along the wells have been
calculated from the travel times using the method of smooth inversion after Lizarralde and Swift (1999). The calculated VSP interval velocities vary between about 2.8 and 5 km/s (Figure 10). Variations within the VSP interval velocity profile show a good correlation to stratigraphy and the dominant lithologies. The VSP interval velocities agree well with the compressional velocities from the sonic log recorded in the lower part of the GrSk3 well.

**4.4    Corridor stacks**

Further processing steps included separation of up- and downgoing wavefields, deconvolution, and transformation to two-way travel time (Table 2). After this, reflections are aligned horizontally and vertical reflection profiles were generated by stacking of the separated upgoing wavefield data over a defined time window after the first arrival (corridor stack). Because of the recording characteristics of the DAS data (see section 4.1), the polarity of
255 the upgoing wavefield data has been reversed, in order to match the polarity convention of conventional geophone data. The polarity convention of the data is European or EAGE normal, i.e. a negative amplitude value (trough) corresponds to an increase in acoustic impedance downwards (Simm and White 2002).

Corridor stacks for GrSk3 and GrSk4 are displayed in Figure 10. The recorded reflections are accurately correlated to depth and can therefore directly be assigned to lithology and other borehole data. The most prominent reflection
events within the corridor stacks occur at the base (reflectors Z1, Z2, Z3) and top (reflectors X1, X2, X3) of the Upper Permian (Zechstein), and within the Middle Triassic (Buntsandstein; reflectors S1, S2).

Larger differences between the corridor stacks are mostly related to intervals where the reflection data is disturbed by residual ringing noise. The slope of this residual noise in the common-source gathers is similar to the slope of reflected upgoing waves, leading to positive superpositions and enhancements in the corridor stack, which cannot
be distinguished from real reflection events.

In the eastern part of the North German Basin, the deepest seismic reflections that can be readily recognized and correlated are at or close to the base of the Zechstein. The reflecting interface Z1 is at the boundary between the Stassfurt Salt and the underlying Stassfurt Anhydrite ("Basalanhydrit"). This "base Zechstein" reflector is used as a marker horizon over the entire Southern Permian Basin area (Doornenbal et al. 2010).

At Groß Schönebeck, the base of the Zechstein is comprised of an 80 – 90 m thick sequence of anhydrite, salt, and carbonate layers, which is underlain by the sediments of the Rotliegend. This interlayered sequence of strata with high impedance contrasts gives rise to several strong and closely spaced reflection bands, which mark the base of Zechstein in the corridor stacks (see Figure 10 and Figure 11).

Reflections within the underlying Rotliegend interval are evident as well, which can now be assigned to individual
sections of the reservoir. The corridor stacks for the Rotliegend reservoir interval are shown in Figure 11, together with well logs and lithology data for both wells. Some of the well logs are unfortunately not available for the lower parts of the wells, especially for GrSk4. Acoustic impedance has been calculated as the product of bulk density and sonic velocity.

The Lower Rotliegend is formed by andesitic volcanic rocks of the Altmark formation. At the depth of the possible
top of the Carboniferous (reflector R8), which was postulated at 4216 mTVDSS for the GrSk3 well, no distinct reflection event is evident in both wells. This is consistent with other regions in the North German Basin, where



the base of the Rotliegend series is essentially non-reflective (Guterch et al. 2010). The transition to the overlying Upper Rotliegend sediments occurs at a depth of 4146 mTVDSS. The corridor stacks show a positive reflection around this depth (reflector H6), which nevertheless has a somewhat different character and a slight offset in depth
of several meters for both wells.

The succession of the Upper Rotliegend sediments starts with the Mirow formation, in which conglomerates are the dominant lithology. The clasts of the rock matrix are lithic fragments of the underlying volcanic rocks. The sediments of the overlying Elbe subgroup are of fluvial and aeolian facies (Gast et al. 1998). Within the lower part, sandstones with good reservoir properties, i.e. high porosities and permeabilities, are occurring within the
Dethlingen formation. Within the wells, the Dethlingen sandstones, which are also known as the Elbe base sandstone, occur as a continuous interval with a thickness of about 100 m, approximately between a depth of about 4000 mTVDSS and 4100 mTVDSS. This interval is characterized by low gamma-ray values, bulk densities and sonic velocities, which corresponds to a low shale content, and increased porosity. It is furthermore marked by a crossover of the bulk density and neutron porosity curves. Bauer et al. (2019) presented an approach to map the
distribution and properties of this sandstone layer based on analysis of seismic attributes of the 3D surface seismic volume.

Above the Dethlingen sandstones, a succession of silt- and mudstones is following. The transition is marked by a change in the log response, with higher values for the gamma-ray, density, and sonic velocity readings, and a separation of the bulk density and neutron porosity curves. This change in density and velocity corresponds to an
overall increase of acoustic impedance. The top of the sandstone interval correlates with a positive amplitude event at a depth of 4010 mTVDSS in both profiles (reflector R3). In GrSk4, another reflection (peak) occurs about 40 m below at a depth of 4051 mTVDSS, which correlates with a step-like decrease of both gamma-ray intensity and sonic velocity. This local change of the log response is not as evident in GrSk3, where only a very weak reflection event occurs at this depth.

The upper interval of the Rotliegend sediments is comprised of an interlayered sequence of silt- and silty mudstones (Hannover formation), with local occurrences of thin-bedded sandy layers. The succession of lithological units and interbeds differs between both wells, which is also reflected by the different character of the corridor stacks within this interval.

The different characteristics of the corridor stacks in the Upper Rotliegend are explained by lithological changes
between the wells. Within the bottom part, the well trajectories have a horizontal distance of up to 475 m, and such lateral changes in lithology are typical for fluvial sediments. The observed character of the reflectors, with low reflectivity and lateral variability, is in line with other regions in the North German Basin, where deeper reflectors within the Rotliegend or Carboniferous commonly cannot be correlated over long distances, because they are of poor quality and often interrupted (Reinhardt 1993).

**5     Summary and conclusions**

Based on this survey, several important new experiences for DAS-VSP acquisition on wireline cable have been gathered. The presented results can be used in support of planning, execution, and evaluation of future surveys of this type.

Common-source gathers of the recorded data are dominated by arrivals of downgoing P waves, upgoing
reflections, and tube waves. One characteristic of the recorded DAS-VSP data is that it is affected by a coherent



noise, which is correlated among neighboring traces. This ringing noise is evident in common-source gathers as a conspicuous zigzag pattern confined to distinct depth intervals, and is occurring in narrow frequency bands. It is influenced by the cable tension and how the cable is aligned with the inner surface of the borehole, depending on changes of the borehole trajectory.

Several tests to determine the influence of the wireline cable tension on the mechanical coupling of the cable to the borehole wall have been performed. The highest signal amplitudes and best overall data quality were found to be achieved under almost full cable tension, and the main part of the data was acquired under these conditions. The results of these tests nevertheless also indicate that a reduction of coherent ringing noise can be achieved by adding cable slack. The interrelation between cable tension and configuration inside the borehole, mechanical coupling to the borehole wall, and recorded signal amplitudes needs further investigation.

After conversion to strain rate, the waveforms and frequency content of the DAS data display a high similarity to vertical component data of a conventional borehole geophone. However, upgoing reflections are recorded with opposite polarity, which confirms the results of earlier studies. The polarity of the reflection data was reversed during later processing, in order to match the polarity of conventional geophone data.

Most of the data has a very good signal-to-noise ratio. Nevertheless, in the GrSk3 well, a sudden reduction of SNR along the deeper part of the profile after the first recording day has been observed. As a larger movement of the cable can be excluded during this time, the cause of this change of acquisition characteristics remains elusive. The ringing noise can be suppressed to a large extent by suitable filtering methods.

From the zero-offset data, accurate time-depth relationships and velocity profiles were derived. The reflectivity
along the boreholes could be mapped with high resolution. The strongest reflections occur at the base and the top of the Zechstein salt complex, and within the Buntsandstein. Nevertheless, in parts the interpretation of the corridor stacks is hampered by residual ringing noise, which is occurring within a short time window after the first break arrivals, and is difficult to be distinguished from true reflection events.

For the Rotliegend reservoir section, the sequence of reflection events in the corridor stacks shows a different
character for both wells overall, which is explained by lateral changes in lithology. But it also displays local similarities: The top of the Dethlingen sandstone interval is marked by a positive reflection event in both wells. This information can be used to identify a related reflector and track the distribution of this reservoir layer in a 3D seismic volume. Processing and interpretation of both 3D VSP and 3D surface seismic data is currently ongoing. The top of the volcanic rocks has a somewhat different response in both wells and no stronger event is obvious at
the postulated top of the Carboniferous. The thickness of the volcanic rocks can therefore not be inferred from individual reflection events in the seismic data alone.

The DAS method has enabled measurements at elevated temperatures up to 150 °C and has led to significant time and cost savings compared to deployment of a conventional geophone chain.

*Data availability.* The seismic survey data will be made available as data publications through GFZ Data Services (https://dataservices.gfz-potsdam.de/portal/).

*Author contributions.* JH and CMK conceptualized the project. JH and MS planned and supervised fieldwork and data acquisition. EM, MS, CMK, and JH performed the seismic data processing and analysis. JH interpreted the
data under discussion with all co-authors, and input from BN on geological well data. JH and EM prepared the manuscript with contributions from all co-authors.



*Competing interests.* The authors declare that they have no conflict of interest. CMK is chief executive editor of SE.

*Acknowledgements.* We gratefully acknowledge the contributions of Ernst Huenges, who led the efforts at the geothermal research platform Groß Schönebeck, and Klaus Bauer for providing major support during funding acquisition and project administration. We are thankful for the smooth cooperation with Schlumberger and DMT GmbH & Co KG during data acquisition, as well as GGL Geophysik und Geotechnik Leipzig GmbH during survey

planning, and Schlumberger and VSProwess Ltd during data processing. Jörg Schrötter, Christian Cunow, and Mathias Poser of GFZ supported fieldwork and acquisition of fiber-optic data. This project has been funded by the German Federal Ministry for Economic Affairs and Energy, grant no. 0324065, and the European Union's Horizon 2020 research and innovation programme under grant agreement no. 691728 (DESTRESS) and 676564 (EPOS-IP).

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





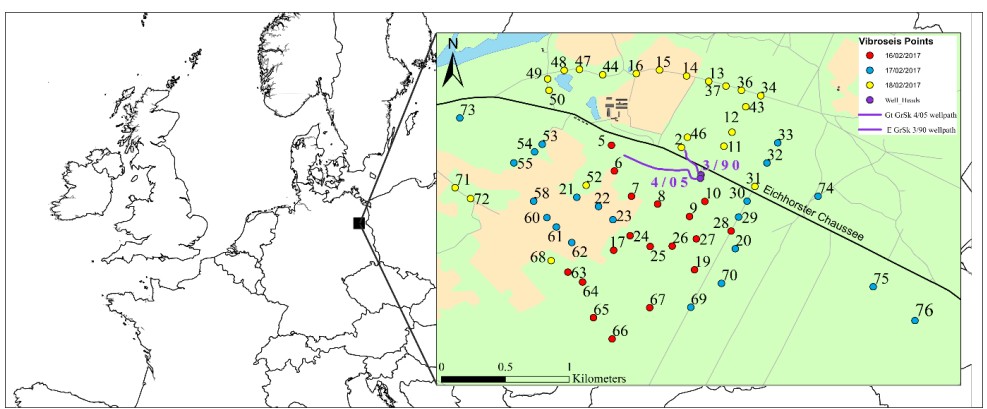

**Figure 1: Overview map of central Europe with location of survey area in NE Germany. Inset shows detail of survey area with VSP source point positions and borehole trajectories of wells E GrSk 3/90 and Gt GrSk 4/05.**

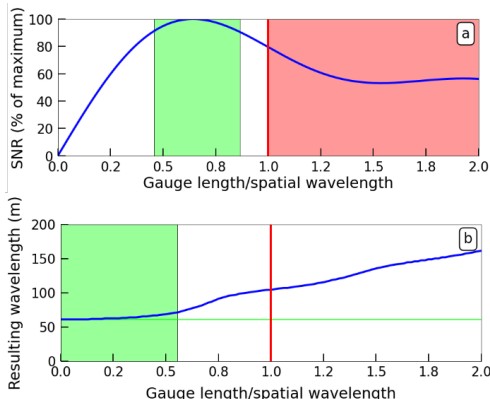

**Figure 2: The signal-to-noise ratio (SNR) (a) and resulting wavelength (b) for different ratios of gauge length to spatial wavelength for a 10-112 Hz Klauder wavelet with a velocity of 4800 m/s. The green boxes indicate the range of ratios where SNR>90% of the maximum, and the resulting wavelength is within 3 m of the actual wavelength. If the gauge length is larger than the spatial wavelength (red box), the wavelet shape is distorted. The optimum conditions satisfying both constraints are found in the region where the two green boxes in (a) and (b) overlap.**





**Figure 3: Selected common-source gathers for well GrSk3 for zero offset, intermediate, and far offset source positions. First column of panels shows data after pre-processing for source positions 10 (a), 25 (b), 66 (g), and 17 (j). Second column (panels b, e, h, and k) shows the data for the same source positions after ringing-noise suppression (Burg adaptive deconvolution) and moderate coherency enhancement. For display, we applied a windowed trace equalisation. The third column (panels c, f, i, and l) shows the signal-to-noise ratio of the data after pre-processing. Colored arrows**
**(exemplary): direct downgoing P wave (light blue), upgoing reflected P-P waves (green), tube wave (magenta), residual noise after application of ringing-noise filter (dark blue).**



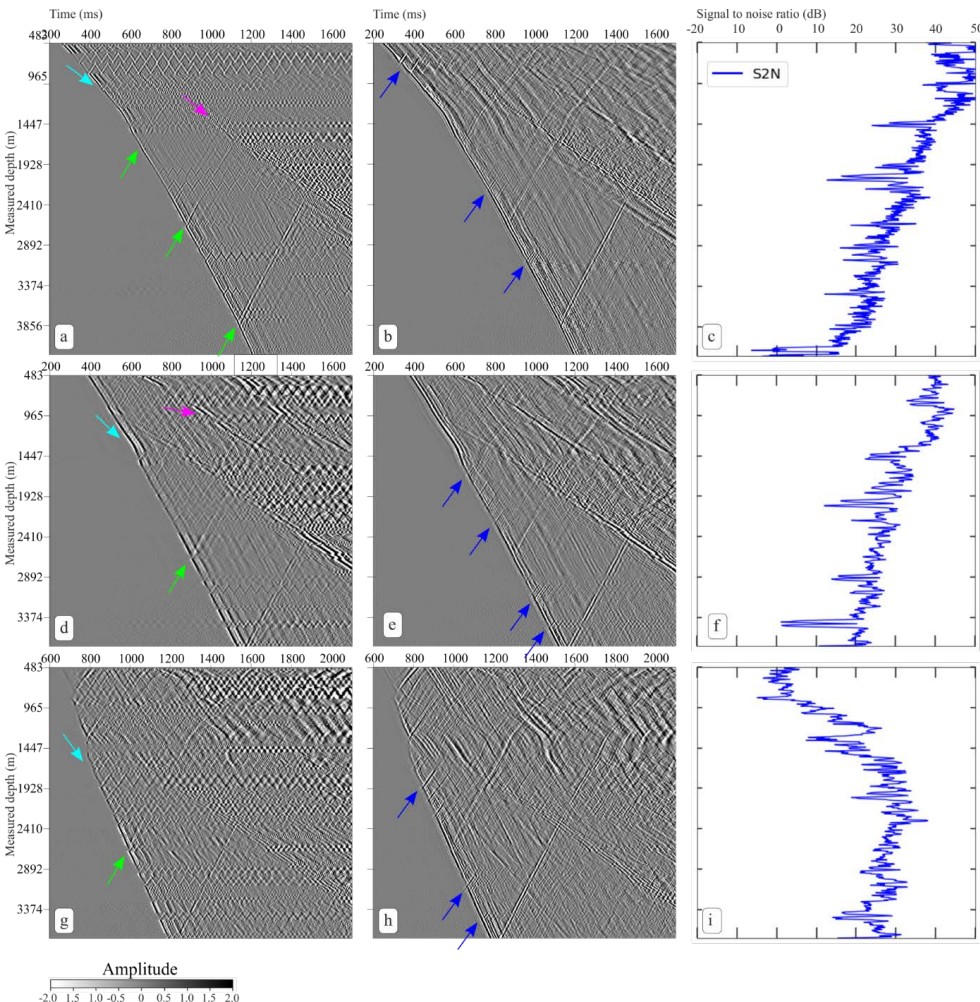

**Figure 4: Selected common-source gathers for well GrSk4 for zero offset, intermediate, and far offset source positions. First column of panels shows data after pre-processing for source positions 10 (a), 25 (b), and 66 (g). Second column (panels b, e, and h) shows the data for the same source positions after ringing-noise suppression (Burg adaptive deconvolution) and moderate coherency enhancement. For display, we applied a windowed trace equalisation. The third column (panels c, f, and i) shows the signal-to-noise ratio of the data after pre-processing. Colored arrows (exemplary): direct downgoing P wave (light blue), upgoing reflected P-P waves (green), tube wave (magenta), residual noise after application of ringing-noise filter (dark blue).**

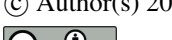



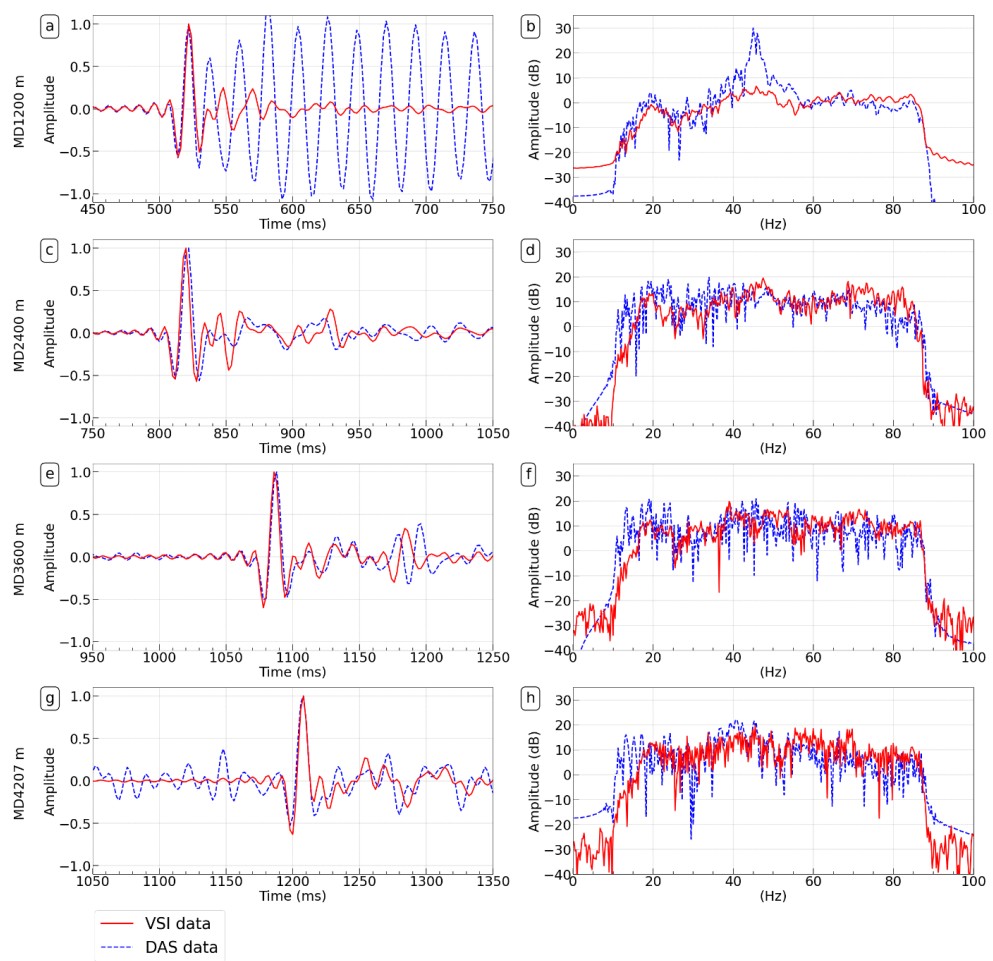

**Figure 5: VSP traces and frequency spectra for borehole geophone data (VSI, red solid line) and DAS strain-rate data (blue dashed line) recorded in well GrSk4 at measured depths of 1200 m (a, b), 2400 m (c, d), 3600 m (e, f), and 4207 m (g, h).**



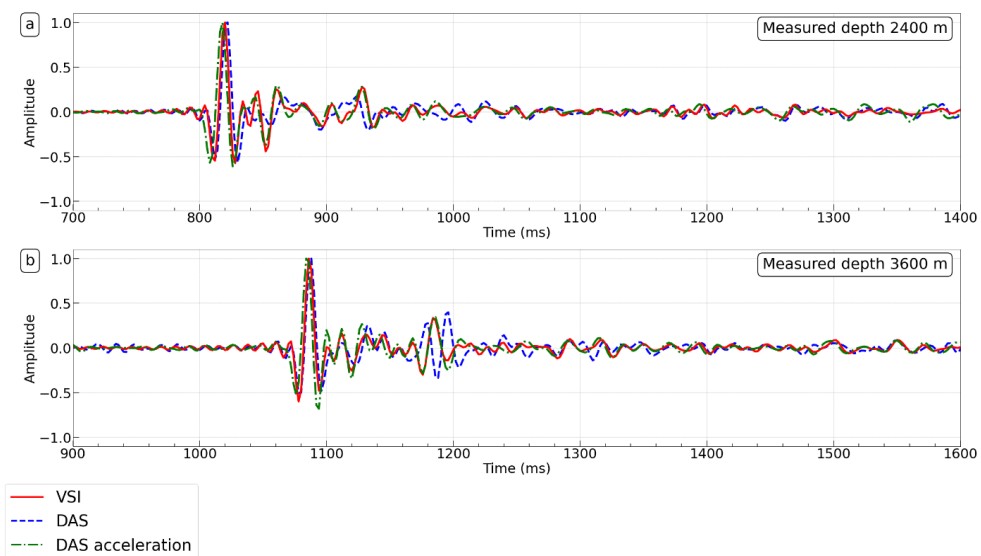

**Figure 6: Comparison of data from the borehole geophone (VSI, red, acceleration), DAS converted to strain rate (blue) and DAS converted to acceleration (green), recorded at measured depths of 2400 m (a) and 3600 m (b) in well GrSk4.**

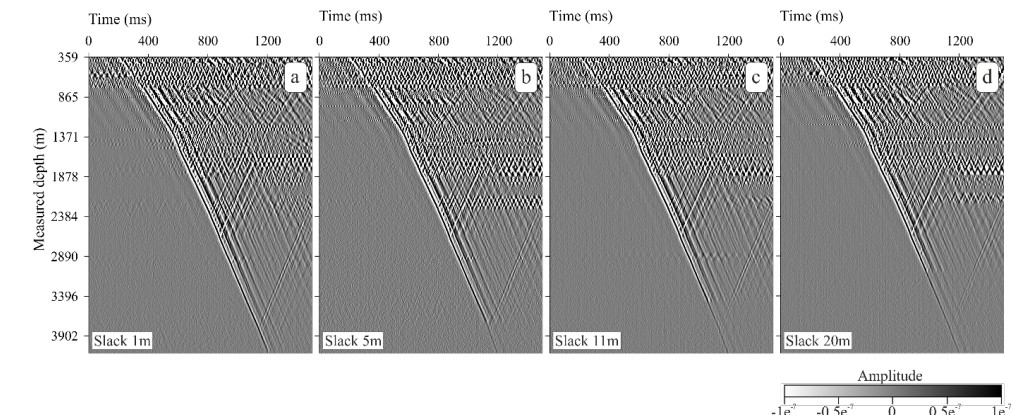

**Figure 7: Comparison of common-source gathers recorded with cable slack of 1 m (a), 5 m (b), 11 m (c), and 20 m (d) for VP10 in well GrSk3.**



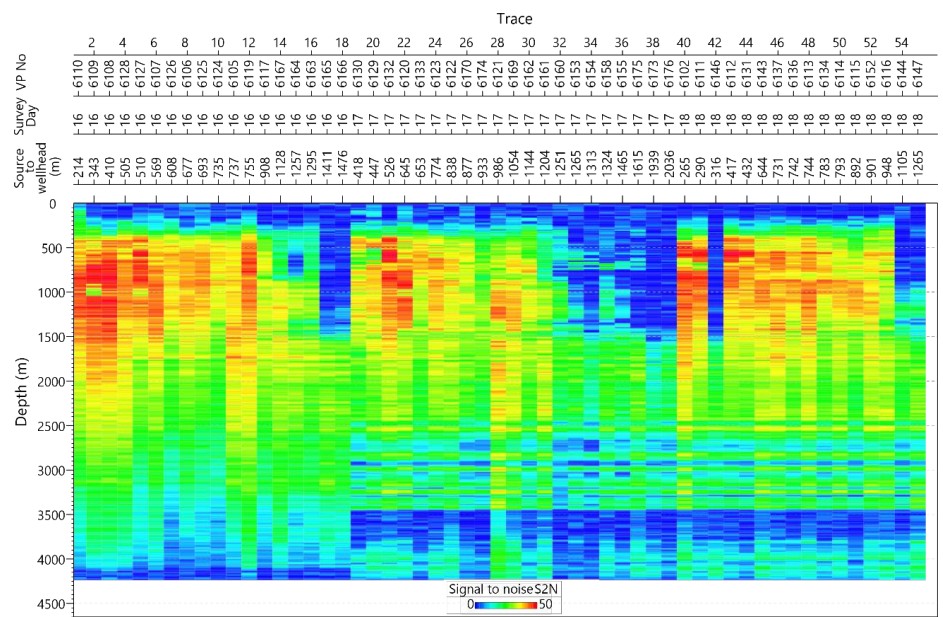

**Figure 8: Signal-to-noise ratio (dB) for DAS-VSP data from well GrSk3.**

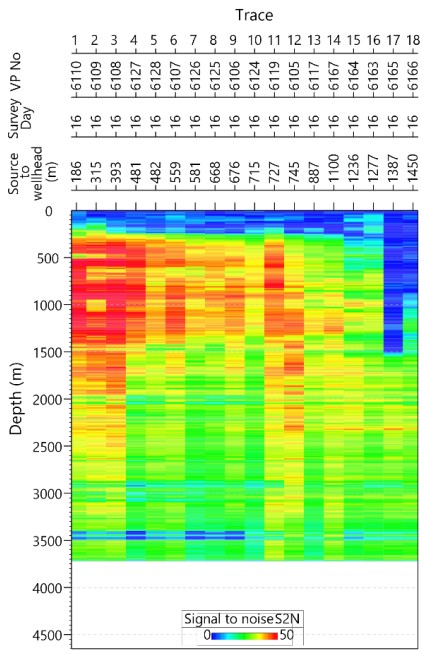

**Figure 9: Signal-to-noise ratio (dB) for DAS-VSP data from well GrSk4.**

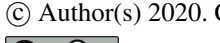



**Figure 10: Vertical one-way travel times (OWT vert), VSP interval velocities (Vint), acoustic log sonic velocities (Vp log), and corridor stacks (CS), together with stratigraphic units, gamma-ray log (GR) and seismic reflectors. TVDSS: True vertical depth below mean sea level.**



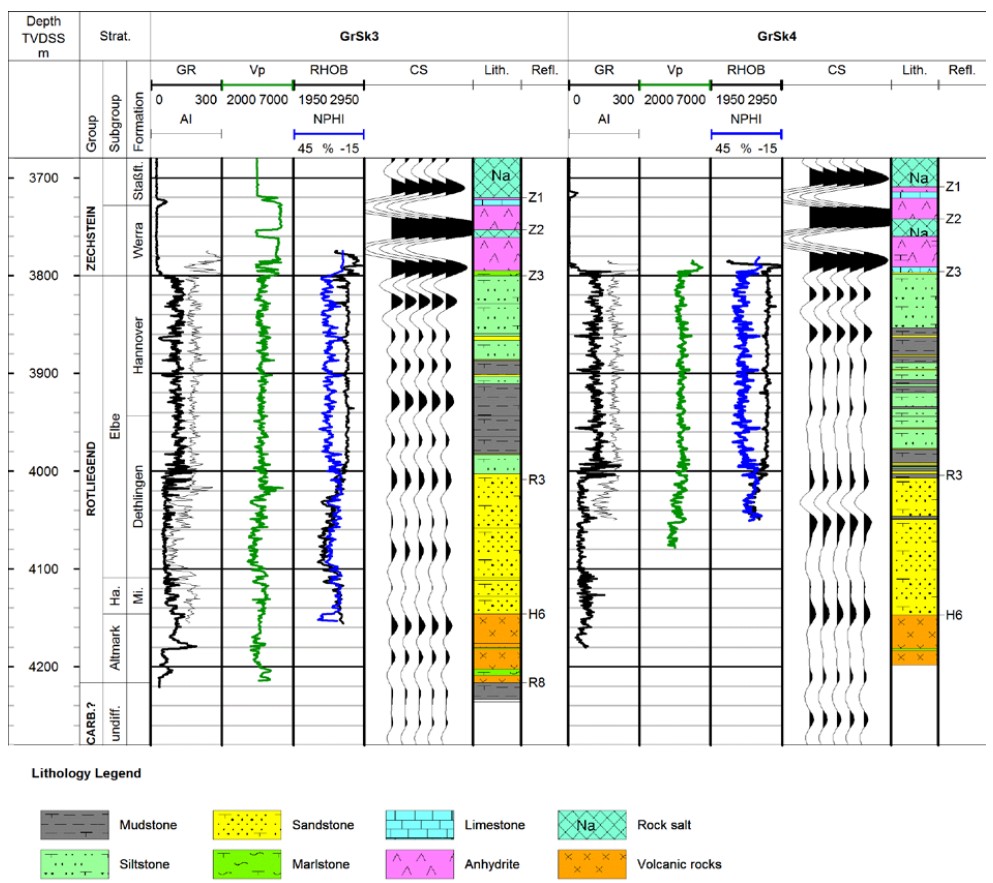

**Figure 11: Corridor stacks (CS) for reservoir interval of the GrSk3 and GrSk4 wells, together with well logs (GR: gamma ray, Vp: sonic velocity, RHOB: bulk density, NPHI: neutron porosity), lithology (Lith.), stratigraphy (Strat.), and seismic reflectors (Refl.). Acoustic impedance (AI) was calculated from bulk density and sonic velocity. TVDSS: True vertical depth below mean sea level.**
