# Peer review of "Wireline distributed acoustic sensing allows 4.2 km-deep vertical seismic profiling of the Rotliegend 150 °C-geothermal reservoir in the North German Basin"

_Solid Earth, 2020_

## Referee Comment (RC1) · Anonymous Referee #1 · 16 Oct 2020

General comments:

The authors present results of DAS-VSP measurements in a wire-line logging approach applied to the Groß Schönebeck site to acquire more information on the structural setting and geometry of the geothermal reservoir. As the authors mention, the use of DAS in a wireline logging approach is a novel application little used until now. Their results demonstrate this approach can be used to retrieve valuable seismic data down to a large depth up to 4256 m, which has not reported before. In addition some unexpected site-specific DAS data characteristics are reported and discussed in more detail. The

final processed an interpreted DAS-VSP results show that the acquired DAS data in combination with other well log-data, contributes to an improved characterization of the target reservoir at significant depths. Overall this is an interesting and relevant paper and is well-written. Therefore I recommend its publication after minor revisions, primarily to add more details regarding the used instruments and processing methods.

Specific comments:

-Page 1, lines 23-25: Consider rephrasing. Do you mean that the top and base of the volcanic rocks can not be inferred from the seismic data due to insufficient reflected energy from these interfaces?

-Page 1, line31-32: Consider rephrasing . For instance: 'This technique allows for rapid seismic data acquisition, because DAS provides continuous point measurements along the cable and therefore does not require vertical repositioning of the cable during VSP campaigns, opposed to conventional geophone borehole strings'

-Page 2, line72-73: Please elaborate on how the source positions were optimized using ray tracing.

-Page 3: line 77-78: Is it correct that this hybrid borehole measurement system includes the interrogator? And could you add the specifications of the fiber-optic cables in table 1?

-Page 3, line 88: I presume that the hDVS is an optical interrogator, please mention this in the text as well.

-Page 4, line 123: Please specify ground units for the different parameters from the equation.

-Page 4, line 133-134: Consider to provide the equation for converting strain to strain rate. Also, does the mentioned 90 $^\circ$ phase shift relate to the 180 $^\circ$ phase shift mentioned in Table 2, or are these not related to each other?

-Page 5, table 5: Please clarify what is meant in the row ''Interval velocities → Correct times to vertical"

-Page 5, line 144-145: Could you comment what the possible cause for the observed zigzag noise pattern is?

-Page 6, line 166-167: Please mention that a comparison between normalized trace amplitudes is made.

-Page 6, line 182: Consider rephrasing, since this is what one would actually expect with DAS data. As for instance Mateeva et al. (2014) state: "Since DAS measures only differential displacement, the polarity of its response is determined by whether a fiber was shortened or elongated over a gauge length, not by the direction of travel of the corresponding seismic wave. "

-Page 6, line 189 with respect to the comparison in Figure 6. The amplitudes of two datatypes seem to be normalized based on their own maximum to [-1,1]. Please mention this. And how do the true unscaled acceleration values actually compare against one another?

-Page 7, line 211-212: And what did their study conclude? Do their modeling outcomes match the observations made in this study? Overall it could be that the effect of the degree of slack is hard to control and largely depends on the well geometry. Maybe certain depth intervals favor from extra slack where coupling is increased, while at other depth intervals the opposite holds depending on trajectory. With this determining optimal slack length could be a matter of trial and error depending partially on well geometry and depth interval/formations of interest for imaging.

-Page 7, line 226-228: Interesting observation. Out of curiosity; did the energy in the noise window increase, or did the energy in the signal window decrease for those intervals?

-Page 8, line 250 regarding section 4.4: Now this section starts with a processing

step and continues with the interpretation of the processed data, although the data processing and interpretation phases should typically be separated. I would therefore recommend to split this in two sections consisting of 4.4 Corridor stack and 4.5 Seismic interpretation.

-Page 10, line 352: Can you comment approximately how much faster DAS-VSP is compared to conventional VSP?

-Page 14, figure 1. Highlight the source positions in the left panel that are further shown in figures 3 and 4 (positions 10, 25, 66 and 17). Please increase the size of the legend in the right panel.

-Page 17, caption of figure 5: Please state that these are normalized amplitudes.

-Figure 18, caption of figure 6. Please state that these are normalized amplitudes. And consider to show and compare non-normalized amplitudes.

Technical corrections:

-Page 8, line 244: Please rephrase the sentence part "For a close to the receiver wells situated zero-offset position, VSP..."

-Page 10, line 398: Regarding the reference to Daley et al. please check the year, because this work seems to date from 2015 instead of 2016.

---

## Referee Comment (RC2) · Ariel Lellouch (Referee) · 20 Oct 2020

Dear authors,

I've had the chance to read manuscript SE-2020-0169, entitled "Vertical seismic profiling with distributed acoustic sensing images the Rotliegend geothermal reservoir in the North German Basin down to 4.2 km depth". I think it shows an exciting practical application that merits to be published. Nonetheless, I have several comments and suggestions, which I detail in the next paragraphs, and hope you will find useful. I'd

be very happy to give the manuscript a second reading for minor fixes after you have addressed these issues.

First of all, I think the focus of the paper could be better. What is fundamentally new in your study is not the VSP processing, which is relatively standard, but the deep deployment of a vertical wireline DAS cable that yields excellent data. This example can have significant implications as old wells in which the fiber was not cemented/tubed will become available for different purposes. The fact that the deployment is in a hot area makes DAS even more attractive. Therefore, I'd suggest a change to the title and to possibly further emphasize this aspect in the text.

Another novel point, which is especially relevant to the DAS community, is the publication of a dataset acquired with the Schlumberger interrogator. One of the main benefits of the hDVS, if I correctly understood the company's claims, is that multiple gauge lengths are acquired during the acquisition step and can later be adjusted during data processing. I suppose there are limitations in what you can show, but it would be nice to see records of different GLs. In addition, you have pronounced changes in velocity, especially at <1500m. In the spirit of your GL optimization, would it be better to use a shorter GL (maybe the 20 m) for the shallower section if the processing workflow allows for it? Besides, the useful Dean et al. (2016) derivation is correct for vertically propagating wavefronts to follow up on that point. In your case, especially for the far offsets, it seems like the apparent slowness is significantly faster, thus commanding longer GL. Would it make sense to have a GL that is offset-dependent as well? I think it is important to discuss the variable GL, because using an average velocity to optimize GL is what one expects for more "standard" interrogators.

There is also a big gap between the acquisition/deployment efforts and the analysis reported here. In the Introduction and Survey design, you set the stage towards 3-D velocity model building/imaging. However, the eventual processing is much more modest – a 1-D velocity analysis with corridor stacks (if I understood correctly). This comment is not a criticism of the work, but I think it would be much better to set reasonable

expectations early on in the paper. Also, I found the velocity model building/processing sections very opaque. Did you use all shots, or just the zero-offset ones, for velocity model building? What is the meaning of a corridor stack for non-zero offsets (the reflection point would be away from the well)? Assuming the 3-D analysis isn't ready, can you show gathers (common-receiver in VSP + direct downgoing travel time correction) to illustrate the azimuthal velocity variations and motivate future 3-D work?

On a related note, the estimated 1-D velocity model has a much lower resolution than the sonic logs, and I think it merits a discussion. The most obvious example is that the high-velocity zone in the top salt (interesting!) around 2500 m, almost 80 m high, only weakly shows in the estimated velocity model. I can understand why it happens to a certain extent. However, looking at the narrower sweep's signatures, it seems to me that you probably have ∼100 Hz with reasonable SNR down to the bottom of the well for most of the survey. So, you have a more than a wavelength that can fit inside the high-velocity zone that I mentioned. I suspect that the low-resolution in the velocity comes from mixing shots in different azimuths and averaging a 3-D structure into the best possible 1-D approximation, but it is only a guess.

Minor technical comments:

1. Sorry for referring to my work, but there have been a few recent deployments of downhole DAS in EGS projects. You can see the studies by Junzo Kasahara (mostly abstracts) and myself (Lellouch et al., 2020, SRL, and another one currently on arXiv). These studies have recognized the benefits of DAS in geothermal reservoirs, and I think they can strengthen your introduction and, eventually, conclusions.

2. It would also be interesting to mention if you picked up any earthquakes during the recording – downhole DAS is much more sensitive than surface arrays.

3. I think Figures 5 and 6 can be combined into one.

4. DAS/geophone comparison - the fact that downhole geophones are actually accelerometers is highly confusing (not your fault. . .). It would be better to clearly state this fact and that the DAS comparison is to the vertical axis of the geophone. By the way, if you follow Egorov 2018 for unit conversions, you can scale each point in the F-K domain by $\omega\text{\textasciicircum}2/k$ and do it all in one pass.

5. Can you show the synthetic (predicted) trace for the logged section? It would be easier to compare with the corridor stacks and see which events are spurious.

6. For non-expert readers, it would be useful to mention why aren't the sonic logs sufficient for your final interpretation. This is not uniquely related to DAS, but to standard VSP as well.

Stay safe and all the best,

Ariel Lellouch, Stanford U.

---

## Author Comment (AC1) · 21 Dec 2020

Dear reviewer,
thank you for your constructive comments and suggestions, which helped us to work out several important aspects of our manuscript more clearly. In the following, we have listed the individual comments, followed by our answers in italic font. After this, the revised manuscript text with the changes highlighted is appended.
Thank you and best regards,
Jan Henninges (on behalf of all authors)

-Page 1, lines 23-25: Consider rephrasing. Do you mean that the top and base of the volcanic rocks can not be inferred from the seismic data due to insufficient reflected energy from these interfaces?

*Response: Yes, the top of the Carboniferous which we had referred to here is equivalent to the base of the volcanic rocks, and we have rephrased the sentence accordingly to make this clearer: "The top of the volcanic rocks has a somewhat different seismic response, and no stronger reflection event is obvious at the postulated base of the volcanic rocks."*

-Page 1, line31-32: Consider rephrasing . For instance: 'This technique allows for rapid seismic data acquisition, because DAS provides continuous point measurements along the cable and therefore does not require vertical repositioning of the cable during VSP campaigns, opposed to conventional geophone borehole strings'

*Response: We have rephrased the sentence accordingly.*

-Page 2, line72-73: Please elaborate on how the source positions were optimized using ray tracing.

*Response: We have added the sentence "Based on the ray tracing, reflection point fold maps for representative layers at target depth and incidence angles of upgoing reflected waves at the sensor cables were calculated and compared for different source point distributions." And the following sentence, lines 73-75, was changed to: "The most suitable source point distribution was then selected, and individual source point locations were further adjusted according to the conditions within the survey area (...)."*

-Page 3: line 77-78: Is it correct that this hybrid borehole measurement system includes the interrogator? And could you add the specifications of the fiber-optic cables in table 1?

*Response: In this case the Schlumberger hDVS interrogator was connected to the hybrid borehole measurement system, as described in the text (p. 3, lines 87-88). The specifications of a hybrid wireline logging cable include many parameters, which would be too much information to be included in Table 1. Within the text on p. 3, we have referred to the publications of Henninges et al. (2011) and Hartog et al. (2014), where descriptions of the individual cables can be found.*

-Page 3, line 88: I presume that the hDVS is an optical interrogator, please mention this in the text as well.

*Response: We have added "optical interrogator" in this sentence.*

-Page 4, line 123: Please specify ground units for the different parameters from the equation.

*Response: We have added the units in parentheses after the definition of the individual*

*parameters for the equations 1 and 2 within the text.*

-Page 4, line 133-134: Consider to provide the equation for converting strain to strain rate. Also, does the mentioned 90 ∘ phase shift relate to the 180 ∘ phase shift mentioned in Table 2, or are these not related to each other?

*Response: The conversion from strain to strain rate by differentiation in time is already described at the specified position in the text (Page 4, line 133-134). So we think that providing an additional equation would be redundant. The 180° phase shift listed in Table 2 has been applied in order to match the polarity of conventional geophone data, as described on p. 8 line 254-256 (we have added a reference to Table 2 at this position for clarity), and is therefore not related to the 90° phase shift referred to here, which results from the differentiation in time.*

-Page 5, table 5: Please clarify what is meant in the row ''Interval velocities → Correct times to vertical''

*Response: The correction to vertical travel times is required because of the deviation of the borehole trajectory, along which the VSP data was recorded, from vertical. It is a standard practice in zero-offset VSP processing, and is briefly described in Section 4.3 "Time depth relationships and interval velocities" on p. 7-8, lines 241-243.*

-Page 5, line 144-145: Could you comment what the possible cause for the observed zigzag noise pattern is?

*Response: A detailed description of our hypothesis for the origin of this "ringing noise" is contained in Section 4 "Results and discussion" on p. 6, lines 171-178.*

-Page 6, line 166-167: Please mention that a comparison between normalized trace amplitudes is made.

*Response: We have added "The recorded amplitudes have been normalized to the absolute maximum first break amplitude of the individual traces." in the captions of Figures 5 and 6.*

-Page 6, line 182: Consider rephrasing, since this is what one would actually expect with DAS data. As for instance Mateeva et al. (2014) state: "Since DAS measures only differential displacement, the polarity of its response is determined by whether a fiber was shortened or elongated over a gauge length, not by the direction of travel of the corresponding seismic wave. "

*Response: We first describe the data we have recorded, and then make references to three earlier studies, including the one of Mateeva et al. (2014), where a similar observation has been made. We think the observed polarity reversal for upgoing reflected waves compared to geophone data is worthwhile to be pointed out, and therefore prefer to keep the text as it is.*

-Page 6, line 189 with respect to the comparison in Figure 6. The amplitudes of two datatypes seem to be normalized based on their own maximum to [-1,1]. Please mention this. And how do the true unscaled acceleration values actually compare against one another?

*Response: Yes, the amplitudes in Figure 6 have also been normalized to the maximum of the first break arrival. We have added "The recorded amplitudes have been normalized to the maximum first break amplitude of the individual traces." in the caption of Figure 6 (also see response to earlier comment on amplitude normalization above, Figure 5).*

-Page 7, line 211-212: And what did their study conclude? Do their modeling outcomes match the observations made in this study? Overall it could be that the effect of the degree of slack is hard to control and largely depends on the well geometry. Maybe certain depth intervals favor

from extra slack where coupling is increased, while at other depth intervals the opposite holds depending on trajectory. With this determining optimal slack length could be a matter of trial and error depending partially on well geometry and depth interval/formations of interest for imaging.

*Response: The study of Schilke et al. (2016) is from the same group as the one from Constantinou et al. (2016), and it describes the numerical simulations referred to in Constantinou et al. (2016) in more detail. The results of this modeling study explain some of the observations, i.e. the constant pitch region, gradually building up from the bottom of the well when further cable slack is introduced. Nevertheless, what the modeling study does not explain, is the decrease of the signal amplitude within this zone, which is observed in our study, similar to the Rittershoffen data set described in Constantinou et al. (2016). So, as noted, it explains parts of the observations but not all, and we have included it as a further reference for the interested reader. With respect to the required slack length, Schilke et al. (2016) note: "The amount of extra cable necessary to be lowered depends on the total length and diameter of the borehole, the dimensions of the cable and its elastic properties, which determine the stiffness."*

-Page 7, line 226-228: Interesting observation. Out of curiosity; did the energy in the noise window increase, or did the energy in the signal window decrease for those intervals?

*Response: Predominantly the energy in the signal window decreased. This is described in the following text already, p. 7, lines 230-231: "The observed signal drop at 3400 m after day 1 seems to be similar to the effect of reduced signal amplitudes observed during the slack test." A common shot gather for a record showing this effect is displayed in Fig. 3, panels j, k, and l. Here we noted, that the number of this VP in the figure caption was erroneous, and we have corrected it accordingly ("17" changed to "76").*

-Page 8, line 250 regarding section 4.4: Now this section starts with a processing step and continues with the interpretation of the processed data, although the data processing and interpretation phases should typically be separated. I would therefore recommend to split this in two sections consisting of 4.4 Corridor stack and 4.5 Seismic interpretation.

*Response: This is correct, and we agree that the data processing and interpretation should usually be separated. In general, we also have followed this practice, and the processing is described in section 3.2 where Table 2 with an overview of the applied processing steps is placed. Nevertheless, as these sections are several pages apart, we think that is helpful for the understanding of the displayed results and our interpretation, if a brief textual description of the related processing steps is given at the beginning of this section here. As this description is rather short (one paragraph with four sentences), we feel that introducing a separate section for this is not justified.*

-Page 10, line 352: Can you comment approximately how much faster DAS-VSP is compared to conventional VSP?

*Response: This is strongly dependent on a number of factors, but in order to give a rough estimate, we have included the following sentence at this position at the end of the text: "Such savings depend on the specific targets and conditions of an individual survey, as well as on the available technologies and performance of the equipment used. But for a VSP survey similar to this study, we would roughly estimate the operational effort to be reduced around a factor of 5 to 10."*

-Page 14, figure 1. Highlight the source positions in the left panel that are further shown in figures 3 and 4 (positions 10, 25, 66 and 17). Please increase the size of the legend in the right panel.

*Response: We have increased the font size of the legend in Figure 1 and marked the respective*

*source point positions with crosses and printed the numbers with bold type. We have added a corresponding explanation in the Figure caption.*

-Page 17, caption of figure 5: Please state that these are normalized amplitudes.

*Response: We have added "The recorded amplitudes have been normalized to the maximum first break amplitude of the individual traces." in the captions of Figures 5 and 6.*

-(Page) 18, caption of figure 6. Please state that these are normalized amplitudes. And consider to show and compare non-normalized amplitudes.

*Response: We have added "The recorded amplitudes have been normalized to the maximum first break amplitude of the individual traces." in the captions of Figures 5 and 6. In our opinion it is not very meaningful to directly compare true amplitudes recorded with different sensors having different sensitivity, characteristics, coupling etc. Trace normalization to the same reference is an appropriate way for a comparison with respect to relative amplitude and S/N ratio.*

Technical corrections:
-Page 8, line 244: Please rephrase the sentence part "For a close to the receiver wells situated zero-offset position, VSP..."

*Response: We have changed this part of the sentence to "For the VP 10 zero-offset position, VSP..."*

-Page 10, line 398: Regarding the reference to Daley et al. please check the year, because this work seems to date from 2015 instead of 2016.

*Response: We have checked the publication date: the work is from 2016.*

[revised manuscript text omitted]

---

## Author Comment (AC2) · 21 Dec 2020

Dear Ariel,
thank you for your constructive comments and suggestions, which helped us to work out several important aspects of our manuscript more clearly. In the following, we have listed the individual comments, followed by our answers in italic font. After this, the revised manuscript text with the changes highlighted is appended.
Thank you and best regards,
Jan Henninges (on behalf of all authors)

First of all, I think the focus of the paper could be better. What is fundamentally new in your study is not the VSP processing, which is relatively standard, but the deep deployment of a vertical wireline DAS cable that yields excellent data. This example can have significant implications as old wells in which the fiber was not cemented/tubed will become available for different purposes. The fact that the deployment is in a hot area makes DAS even more attractive. Therefore, I'd suggest a change to the title and to possibly further emphasize this aspect in the text.

*Response: We agree and have modified the title to "Wireline distributed acoustic sensing allows 4.2 km-deep vertical seismic profiling of the Rotliegend 150°C-geothermal reservoir in the North German Basin", as well as several parts of the abstract accordingly, in order to work out these points more clearly.*

Another novel point, which is especially relevant to the DAS community, is the publication of a dataset acquired with the Schlumberger interrogator. One of the main benefits of the hDVS, if I correctly understood the company's claims, is that multiple gauge lengths are acquired during the acquisition step and can later be adjusted during data processing. I suppose there are limitations in what you can show, but it would be nice to see records of different GLs.

*Response: It is true that with Schlumberger's hDVS technology, the gauge length can be varied, both during a survey and after recording during data processing. We used this feature and recorded data sets with different gauge lengths during the start-up test. Based on this, we had chosen a gauge length of 20 m for recording in the field, as described in section 2. Then, after derivation of the travel times and interval velocities from the recorded data, we performed the gauge length optimization procedure of Dean et al. (2017) using the average velocity of the target interval, as described in section 3.1. This resulted in an optimum gauge length of 40 m, for which the whole data set was then reprocessed. We will include data recorded or processed with different gauge length in the data publication/supplement to this paper, which is currently under preparation: Henninges J., Martuganova E., Stiller M., Norden B., and Krawczyk C.M.: DAS-VSP data from the Feb. 2017 survey at the Groß Schönebeck site, Germany, GFZ Data Services, doi:10.5880/GFZ.4.8.2021.001, in review. We have added the reference to the data publication under section "Data availability" and the reference list.*

In addition, you have pronounced changes in velocity, especially at <1500m. In the spirit of your GL optimization, would it be better to use a shorter GL (maybe the 20 m) for the shallower section if the processing workflow allows for it?

*Response: Yes, for processing of the shallower intervals this would be beneficial, as described e.g. in Dean et al. (2017). But as the focus of our current study is predominantly on the deeper Rotliegend reservoir section, such a zoned processing approach was not applied here. We have therefore added the following sentence at the end of section 3.1: "It would also be possible to apply a depth-dependent gauge length optimization, as suggested by Dean et al. (2017), by taking local variations of velocity and frequency content into account. This was nevertheless not performed in the current study, because the focus here is predominantly on the deeper*

*Rotliegend reservoir section only.*

Besides, the useful Dean et al. (2016) derivation is correct for vertically propagating wavefronts to follow up on that point. In your case, especially for the far offsets, it seems like the apparent slowness is significantly faster, thus commanding longer GL. Would it make sense to have a GL that is offset-dependent as well? I think it is important to discuss the variable GL, because using an average velocity to optimize GL is what one expects for more "standard" interrogators.

*Response: True, but again, as the focus of our current study is on the deeper Rotliegend reservoir section only, where larger variations of velocity do not occur, as well as on the zero-offset data sets, we feel that it would be somewhat outside the scope of this paper to enter into a deeper discussion on variable gauge lengths. But as noted above, we have included a note on the importance and our approach used here within the text.*

There is also a big gap between the acquisition/deployment efforts and the analysis reported here. In the Introduction and Survey design, you set the stage towards 3-D velocity model building/imaging. However, the eventual processing is much more modest – a 1-D velocity analysis with corridor stacks (if I understood correctly). This comment is not a criticism of the work, but I think it would be much better to set reasonable expectations early on in the paper.

*Response: To make this clearer we have added the following sentences at the end of section 1 Introduction: "In the following, the survey design and data acquisition, the overall characteristics of the acquired data, as well as the data processing and evaluation for a zero-offset source position are presented. Specific processing and interpretation of a 3D-VSP seismic cube will be the subject of a separate publication."*

Also, I found the velocity model building/processing sections very opaque. Did you use all shots, or just the zero-offset ones, for velocity model building? What is the meaning of a corridor stack for non-zero offsets (the re- flection point would be away from the well)?

*Response: The building of the velocity model was intentionally only described briefly here, as it was only used for correction of the vertical travel times, as described at the beginning of section 4.3. The velocity model building was part of the 3D-VSP imaging, which will be described in a separate publication, as noted above. The corridor stacks were of course only prepared for the VP10 zero-offset source position. In order to make this clearer we have added "applied to the data from the VP 10 zero-offset position" to the first sentence at the beginning of section 4.4.*

Assuming the 3-D analysis isn't ready, can you show gathers (common-receiver in VSP + direct downgoing travel time correction) to illustrate the azimuthal velocity variations and motivate future 3-D work?

*Response: In contrast to a common-source gather (one source point recorded by a linear receiver line) it is hardly possible to display a common-receiver gather containing spatially distributed source positions with different offsets and azimuths recorded at one receiver point. We don't believe that an azimuthal velocity analysis can be performed by this.*

On a related note, the estimated 1-D velocity model has a much lower resolution than the sonic logs, and I think it merits a discussion. The most obvious example is that the high-velocity zone in the top salt (interesting!) around 2500 m, almost 80 m high, only weakly shows in the estimated velocity model. I can understand why it happens to a certain extent. However, looking at the narrower sweep's signatures, it seems to me that you probably have ~100 Hz with reasonable SNR down to the bottom of the well for most of the survey. So, you have a more than a wavelength that can fit inside the high-velocity zone that I mentioned. I suspect that the low-resolution in the velocity comes from mixing shots in different azimuths and averaging a 3-D structure into the best possible 1-D approximation, but it is only a guess.

*Response: Of course the sonic log has a much higher spatial resolution due to the higher source frequency the tools are using, which usually reaches up to several kHz. But your assumption that the high velocity layer at around 2.500 m depth (a 50 m thick anhydrite layer, with the X2 and X3 reflectors at its top and base, respectively) should be resolvable in the VSP data is nevertheless correct. This layer, as well as the other two high-velocity anhydrite layers at the base of the Zechstein interval, intentionally appear strongly smoothed in the VSP interval velocity profiles, because these were calculated using the method of smooth inversion after Lizarralde & Swift (1999), as described in section 4.3 / line 245 within the text. For further explanation of this method, we have added the following text: "Here a damped least-squares inversion of VSP travel times is applied, which reduces the influence of arrival-time picking errors for closely spaced sampling points, and seeks to result in a smooth velocity/depth profile. In our study, a 1.1 ms residual of the travel times has been allowed for." In the last sentence of this section on the comparison with the sonic log data, we further added: "Taking into account the desired smooting of the profiles resulting from the applied computation method, (...)".*

Minor technical comments:
Sorry for referring to my work, but there have been a few recent deployments of downhole DAS in EGS projects. You can see the studies by Junzo Kasahara (mostly abstracts) and myself (Lellouch et al., 2020, SRL, and another one currently on arXiv). These studies have recognized the benefits of DAS in geothermal reservoirs, and I think they can strengthen your introduction and, eventually, conclusions.

*Response: Such studies on using DAS for microseismic monitoring are indeed strongly relevant for our study. We have included the following sentence including this and some further references in the introduction: "There is also a growing number of studies applying DAS for microseismic monitoring during hydraulic stimulation (e.g. Molteni et al. 2017, Karrenbach et al. 2017), also including EGS reservoirs (Lellouch et al. 2020)." The following three references were added:*
*Karrenbach, M., Kahn, D., Cole, S., Ridge, A., Boone, K., Rich, J., Silver, K., and Langton, D.: Hydraulic-fracturing-induced strain and microseismic using in situ distributed fiber-optic sensing, Leading Edge, 36, 837-844, 10.1190/tle36100837.1, 2017.*
*Lellouch, A., Lindsey, N. J., Ellsworth, W. L., and Biondi, B. L.: Comparison between Distributed Acoustic Sensing and Geophones: Downhole Microseismic Monitoring of the FORGE Geothermal Experiment, Seismol. Res. Lett., 91, 3256-3268, 10.1785/0220200149, 2020.*
*Molteni, D., Williams, M. J., and Wilson, C.: Detecting microseismicity using distributed vibration, First Break, 35, 51-55, 2017.*
It would also be interesting to mention if you picked up any earthquakes during the recording – downhole DAS is much more sensitive than surface arrays.

*Response: We did not notice any signs of earthquakes during the processing of the data. The natural seismic activity in the study area is very low.*

I think Figures 5 and 6 can be combined into one.

*Response: We deliberately plotted this data in two separate graphs, in order to have a clear visualization of the individual traces. We think that the plots would become overloaded when combining the two figures into one, therefore we prefer to keep them separate.*

DAS/geophone comparison - the fact that downhole geophones are actually accelerometers is highly confusing (not your fault. . .). It would be better to clearly state this fact and that the DAS comparison is to the vertical axis of the geophone. By the way, if you follow Egorov 2018 for unit conversions, you can scale each point in the F-K domain by $\omega^2/k$ and do it all in one pass.

*Response: The fact that the borehole geophone is an accelerometer is stated within the text in lines 86 and 166, where it is also stated, that the vertical component is displayed, as well in the caption of Fig. 6. We have now also added this information in the caption of Figure 5: "The*

*borehole geophone is a three-component accelerometer, and the vertical component parallel to the tool/borehole axis is displayed."* With respect to the data conversion according to Egorov et al. (2018), yes, you are right, thank you for this notice. We decided to split the conversion process into these two sub-steps for easier comprehension. Technically it is the same procedure.

Can you show the synthetic (predicted) trace for the logged section? It would be easier to compare with the corridor stacks and see which events are spurious.

*Response: We had calculated different synthetics with variation of the input wavelets, spatial sampling, etc., in the course of our analysis. But as the logging data exists only for certain intervals and the results were not fully conclusive, we have not included these here.*

For non-expert readers, it would be useful to mention why aren't the sonic logs sufficient for your final interpretation. This is not uniquely related to DAS, but to standard VSP as well.

*Response: In the revised abstract and introduction we explain that the DAS-VSP survey was performed in order to gain more detailed information on the structural setting and geometry of the reservoir, and that we derived time-depth relationships, interval velocities, and corridor stacks from the data. Clearly, most of this information is not provided by a sonic log. So this should be clear now, even to non-expert readers.*

[revised manuscript text omitted]

---

## Author Response (AR2)

Author response to comments on revised manuscript "Wireline distributed acoustic sensing allows 4.2 km-deep vertical seismic profiling of the Rotliegend 150°C-geothermal reservoir in the North German Basin" by Jan Henninges et al. from the topical editor Zack Spica

Dear editor,

thank you for your comments and suggestions to our revised manuscript, which we gratefully follow. In the following, we have listed the individual comments, followed by our answers in italic font, including the response to the comments from the referee. The revised manuscript text with all changes highlighted is provided as a separate document, as requested by the SE upload form.

Thank you and best regards,

Jan Henninges (on behalf of all authors)

1) p1.l30: "because DAS provides continuous point measurements along the cable [...]". DAS does not provide point measurements but an integrated measure of strain (rate) over a gauge length. For this reason, a DAS channel and a geophone, even if collocated, will never provide the exact same seismogram.

*Response: Our formulation was indeed a bit misleading, and we have changed "continuous point measurements" to "continuous measurements".*

2) Fig. 2b: can you clarify what are the horizontal green line and the vertical red line?

*Response: The horizontal green line marks the actual wavelength, and the vertical red line a gauge length to spatial wavelength ratio of 1, above which the wavelet shape is distorted. We have added this information to the caption of Figure 2.*

3) I could not find your data either. Importantly, please provide a direct link to the data, not a link to the data portal. The doi provided is not recognized.

*Response (also see response to comment no. 4 from the reviewer): In the revised manuscript we have included a direct link to the data publication with the doi 10.5880/GFZ.4.8.2021.001. This doi number is currently reserved but not activated yet, because the data publication is still in preparation and under review. A preview of the metadata is nevertheless already available under the following link:*
*https://dataservices.gfz-potsdam.de/panmetaworks/review/3ea650a18f1bfd5993ac9d33d6a47fb6848133d64b1969501c2921ac7d2a51da/*

*The data publication will be finalized in the coming days and the doi will then be activated. We will update the reference to the data publication, which currently is Henninges et al., in review, during the final print proof stage.*

4) Please update the "in review" references if there is any change since the last submission.

*Response: There is no change of the status of the two concerned references "in review" since the last submission. But we expect that we can do this in the final print proof stage.*

Author response to report # 1 on revised manuscript "Wireline distributed acoustic sensing allows 4.2 km-deep vertical seismic profiling of the Rotliegend 150°C-geothermal reservoir in the North German Basin" by Jan Henninges et al. from referee #2 Ariel Lellouch

Dear Ariel,

thank you for your additional comments and suggestions to our revised manuscript, which are again very constructive and helped us to further improve the manuscript. In the following, we have listed the individual comments, followed by our answers in italic font. The revised manuscript text with all changes highlighted is provided as a separate document, as requested by the SE upload form.

Thank you and best regards,

Jan Henninges (on behalf of all authors)

1) I think it would be useful to mention the multi-gauge length recording property of hDVS.

*Response: We have now included this information into the corresponding sentence in section 2, line 102, as follows: "After testing of several different gauge length values (see Henninges et al. in review), which can be varied with the hDVS interrogator, a gauge length value of 20 m was selected for online DAS data processing during recording."*

2) I don't know about Solid Earth's policy, but it is worth checking if papers under review/preparation can be referenced.

*Response (also see comment no. 4 by the topical editor): The SE policy for references like these is: >>Works "submitted to", "in preparation", "in review", or only available as preprint should also be included in the reference list.<< We expect to be able to update the two concerned references in the final print proof stage.*

3) I'd be very interested in reading alternative theories on why the SNR patterns changed throughout the experiment (if they exist).

*Response: We would also be very interested in such theories. But we are not aware that similar observations, or theories which could explain them, have been described elsewhere until now.*

4) The link to the data isn't working for me.

*Response (also see comment no. 3 by the topical editor): In the revised manuscript we have included a direct link to the data publication with the doi 10.5880/GFZ.4.8.2021.001. This doi number is currently reserved but not activated yet, because the data publication is still in preparation and under review. A preview of the metadata is nevertheless already available under the following link:*
*[https://dataservices.gfz-potsdam.de/panmetaworks/review/3ea650a18f1bfd5993ac9d33d6a47fb6848133d64b1969501c2921ac7d2a51da/](https://dataservices.gfz-potsdam.de/panmetaworks/review/3ea650a18f1bfd5993ac9d33d6a47fb6848133d64b1969501c2921ac7d2a51da/)*

*The data publication will be finalized in the coming days and the doi will then be activated. We will update the reference to the data publication, which currently is Henninges et al., in review, during the final print proof stage.*

Line 32 >> Stay consistent between geophone chain (abstract) and string (here). I prefer string

*Response: For consistency, we have now used the expression "borehole geophone string" throughout the text.*

Line 47 >> Within *the* two research wells existing at the site (since you already mentioned them)

*Response: We have changed this part of the sentence to "(...) within the GrSk3 and GrSk4 wells."*

Line 50 >> I'd explicitly state that this formation is the target of the study.

*Response: We have modified the corresponding section to: "(...) and to image structural elements within the reservoir interval of the Rotliegend at 4200 m depth in the vicinity of the boreholes with higher resolution in three dimensions. The imaging of structures in the target reservoir interval is a special challenge, (...)"*

Line 62 >> "For this wireline deployment method nevertheless only very few experiences exist until now" – I think the phrasing is slightly off, consider rewriting

*Response: We think that that our statement, that only very few experiences for wireline deployments of fiber-optic sensor cables (sensu stricto) exist until now is justified, given the handful of existing publications on the subject (most of them cited in our manuscript), and would prefer to leave the formulation as it is.*

Line 67 >> I think "as well" should be "and" because of the comma preceding it

*Response: We have changed "as well" to "and".*

Line 95 (and others) >> I think that "DAS data were" is better than "DAS data was"

*Response: Historically, the use of plural with "data" is correct. But today the usage of "data" as a mass noun with singular verb is widely accepted, see e.g. Oxford Dictionary of English.*

Line 114 >> "mainly data for well GrSk3 could only be recorded during this time." – remove "only" or rephrase

*Response: We have rephrased this to "mainly only data for well GrSk3 could be recorded during this time."*

Line 127 >> "signal-to-noise ratio, while keeping" – unnecessary comma

*Response: We have deleted the comma.*

Line 131 >> Emphasize that it is the apparent velocity (ZO case)

*Response: We have changed "velocity" to "(apparent) velocity".*

Line 174 >> "to be distinguished" -> to distinguish

*Response: We think that both formulations are be possible in this case, and we prefer to keep "to be distinguished".*

Line 186 >> "acoustic receivers" is ambiguous, maybe "recording elements" instead?

*Response: We have changed "(...) the ringing noise in the DAS data is related to the different deployment methods of the acoustic receivers." to "(...) the ringing noise in the DAS data is related to the deployment method of the DAS sensor cable."*

Line 225 >> "The energy of signal and noise was computed" – should be in plural form

*Response: We have changed "energy" to plural: "The energies of signal and noise were computed (...)".*

Line 282 >> "and correlated are at or close to" – the "are" is redundant

*Response: No, this is not correct, "are" is the predicate of the sentence and is required.*

Line 331 >> "Based on this survey, several important new experiences" – maybe "lessons" works better?

*Response: This would be an alternative formulation as well, but we prefer to keep "experiences".*

Line 365 >> can therefore not -> can not therefore

*Response: The proposed formulation seems strange. We prefer to keep the original one.*